# Dopamine Neuron Diversity: Recent Advances and Current Challenges in Human Stem Cell Models and Single Cell Sequencing

**DOI:** 10.3390/cells10061366

**Published:** 2021-06-01

**Authors:** Alessandro Fiorenzano, Edoardo Sozzi, Malin Parmar, Petter Storm

**Affiliations:** Developmental and Regenerative Neurobiology, Wallenberg Neuroscience Center, and Lund Stem Cell Centre, Department of Experimental Medical Science, Lund University, 22184 Lund, Sweden; edoardo.sozzi@med.lu.se (E.S.); malin.parmar@med.lu.se (M.P.); petter.storm@med.lu.se (P.S.)

**Keywords:** human pluripotent stem cells, dopamine neuron differentiation, Parkinson’s disease, human brain organoids, cell transplantation, single cell RNA sequencing

## Abstract

Human midbrain dopamine (DA) neurons are a heterogeneous group of cells that share a common neurotransmitter phenotype and are in close anatomical proximity but display different functions, sensitivity to degeneration, and axonal innervation targets. The A9 DA neuron subtype controls motor function and is primarily degenerated in Parkinson’s disease (PD), whereas A10 neurons are largely unaffected by the condition, and their dysfunction is associated with neuropsychiatric disorders. Currently, DA neurons can only be reliably classified on the basis of topographical features, including anatomical location in the midbrain and projection targets in the forebrain. No systematic molecular classification at the genome-wide level has been proposed to date. Although many years of scientific efforts in embryonic and adult mouse brain have positioned us to better understand the complexity of DA neuron biology, many biological phenomena specific to humans are not amenable to being reproduced in animal models. The establishment of human cell-based systems combined with advanced computational single-cell transcriptomics holds great promise for decoding the mechanisms underlying maturation and diversification of human DA neurons, and linking their molecular heterogeneity to functions in the midbrain. Human pluripotent stem cells have emerged as a useful tool to recapitulate key molecular features of mature DA neuron subtypes. Here, we review some of the most recent advances and discuss the current challenges in using stem cells, to model human DA biology. We also describe how single cell RNA sequencing may provide key insights into the molecular programs driving DA progenitor specification into mature DA neuron subtypes. Exploiting the state-of-the-art approaches will lead to a better understanding of stem cell-derived DA neurons and their use in disease modeling and regenerative medicine.

## 1. Introduction

Dopamine (DA) neurons in the ventral midbrain (VM) constitute a heterogeneous group of cells with different anatomical locations, physiological properties, and projection patterns. These cells are involved in a broad spectrum of cerebral functions associated with voluntary movement, as well as with cognitive and emotive tasks [1,2].

DA neurons are traditionally divided into three subtypes. A9 neurons are located in the substantia nigra pars compacta (SNpc), A10 in the ventral tegmental (VTA) area, and A8 in the retrorubral field. As well as displaying different functions and axonal innervations, they exhibit heterogeneous susceptibility to disease processes and show fundamental differences in vulnerability to cell death in Parkinson’s disease (PD). Each cell group forms specific connections and projects to distinct anatomical target areas of the central nervous system (CNS), establishing separately controlled functional networks [3,4,5]. A8 and A10 neurons innervate the ventral striatum, nucleus accumbens, septum, and the prefrontal cortex via the mesolimbic pathway, and are mainly involved in controlling emotional behavior and motivation. A9 neurons project to the striatum, forming the nigrostriatal pathway that regulates motor function. They also have different functional and projection patterns, as compared to the other subtypes. A9 neurons in humans and other primates also show accumulation of the neuromelanin pigment. A9 neurons are primarily degenerated in PD, making them the subject of more extensive studies [6,7,8,9,10].

Given the key role of A9 cells in PD, the generation of this subtype of DA neurons from stem cell sources is an area of intense investigation ultimately aimed at exploiting their use in cell-based replacement treatment. For decades, neuroscientists have attempted to identify selective markers expressed in these subpopulations, in order to dissect the complexity of DA regulatory networks and design more effective therapeutic strategies [11,12]. The inaccessibility of fetal and adult human brain tissue makes it difficult to elucidate the relation between histological assessments and the heterogeneity of DA neurons at the molecular level. The compilation of a comprehensive dataset linking the molecular diversity of DA neurons with their function and anatomical innervation target would require a systematic genome-wide molecular classification at single-cell resolution [13,14].

The advent of single cell sequencing technologies has provided unprecedented insights into DA subtypes and uncovered an unexpectedly high heterogeneity, even within anatomically defined DA subgroups. Such approaches have already been used in the adult mouse brain to unbiasedly catalog DA neurons, based on their gene expression profiles. However, the question of whether a similar diversity exists in the human midbrain and whether molecularly distinct DA subtypes correspond to innervation target regions and the traditional classification based on anatomical landmarks, remains completely unexplored [3,15,16].

The ability to recreate human neurons from human pluripotent stem cells (hPSCs) opens up exciting opportunities to study human neurogenesis and understand mechanisms and treatments for brain disease(s). It also provides access to a renewable source of cells potentially suitable for therapeutic applications, including drug screening and cell-based therapy (Figure 1) [17,18]. The generation of human tissues in vitro, combined with the use of human cells in xenograft transplantation models and sophisticated transcriptomics technologies, is ushering in the new era of “human” biology. Breaking down the intricate regulatory system controlling DA neuron subtype specification into its individual layers will provide crucial new insights into the transcription factors and molecular cues that specifically drive the diversity of human DA neurons [19,20,21].

This review focuses on recent advances and current challenges in strategies used to study the maturation and diversification of human neurons and to model the complexity of human brain tissue, which is virtually inaccessible to research for practical and ethical reasons. We also describe the importance of computational single-cell analyses in this setting and their application in tracing developmental trajectories and exploring the morphological and functional properties of mature human DA neurons.

## 2. Cell-Based Human Models In Vitro and after Transplantation In Vivo 

Our understanding of the molecular mechanisms controlling human brain development is mainly restricted by the tremendous heterogeneity in cell composition, cytoarchitecture, and physiology of the brain tissue. The inaccessibility of the human brain makes it extremely challenging to recapitulate its development, function, and dysfunction. Much of our current knowledge is derived from the histological analysis of post-mortem pathological tissue, since the use of endogenous brain tissue from humans and primates is severely limited for ethical and practical reasons. Research efforts are further complicated by difficulties associated with the expansion and genetic manipulation of human tissue [22,23,24,25]. While advancements have been made by focusing largely on rodents, any comparison between human physiology and mouse models is inherently impeded by species differences. The study of human CNS diseases in animal model systems has therefore proved inadequate [26,27]. Investigations into neurodegenerative diseases such as PD and Alzheimer’s using animal modeling are even more problematic, since the etiology of these disorders is usually complex and diverse and is further complicated by individual risk variants, distinct genetic and epigenetic backgrounds, and unknown environmental triggers [28,29].

The use of hPSCs instead of animal models holds the promise of recreating key molecular and architectural features of the human brain. Similarly, exploiting induced pluripotent stem cells (iPSCs) reprogrammed from healthy/diseased individuals could be used to recapitulate neurodevelopmental and neurodegenerative disorders, in order to investigate the causal relationship between disease phenotype and patient-specific genetic patterns (Figure 1) [30,31]. In the following subsections we discuss the use of stem cell models to reproduce the maturation and diversification of human DA neurons.

### 2.1. 2D Culture

In the last 20 years, the most widely used differentiation protocols have all been based on adherent 2D cellular cultures, and a variety of approaches driving stem cell commitment toward neuron differentiation are currently available [32,33]. Although a 2D configuration limits cell growth to a monolayer and is reductionist in nature, 2D cell culture has the great advantage of generating homogeneous populations of differentiated cells with high purity, and of enabling uniform accessibility to patterning factors and gas exchange [20,34]. The ability to create homogeneously differentiated cells that are easy to genome edit makes 2D culture highly suitable for cell replacement therapies. Monolayer conditions can also be readily adapted to small culture sizes for scalable protocols that are useful in high-throughput drug screenings [35,36]. The exploitation of 2D culture properties has led to the redesign and optimization of hPSC differentiation protocols. Initial approaches to the generation of human DA neurons produced incorrectly specified tyrosine hydroxylase cells, which although capable of releasing dopamine, displayed poor survival after transplantation and gave rise to undesirable derivative cell types [37,38]. 

Redefining stem cell DA neuron differentiation overcame this seemingly intractable problem and profoundly altered PD research. A breakthrough in the optimization of DA differentiation protocols occurred with the publication of initial insights into the cellular origin of DA neurons. Interestingly, the authors showed that DA neurons are not derived from PAX6+ neuroepithelial progenitors, but from the FOXA2+/LMX1A+ domain [39,40]. This discovery resulted in the emergence of refined 2D differentiation protocols aimed at generating DA neurons from a VM floor plate rather than PAX6+ intermediate progenitors. To date, most protocols for driving hPSCs into DA neurons share the same patterning factors. Dual SMAD inhibitors blocking TGFb and BMP signaling, combined with the ventralizing factor SHH and the caudalizing factor CHIR99021 for WNT pathway activation, are used for efficient specification of VM progenitors expressing a unique combination of markers found only in the midbrain floor plate, including FOXA2, LMX1A, EN1, and CORIN [17,41]. hPSC-derived DA progenitors can further differentiate into functional DA neurons with electrophysiological properties, and evidence indicates that they can also survive and mature after intracerebral transplantation in a rat model of PD, giving rise to DA-rich grafts that are able to innervate the host brain, leading to functional recovery [20,33].

Direct neural conversion has emerged as an alternative strategy to recapitulate highly mature neurons in a dish for both disease modeling and cellular repair. Somatic human cells can be directly converted to distinct types of neurons via a combined expression of specific factors. Cells generated using this approach do not undergo an intermediate pluripotent state, attenuating the risk of tumorigenesis, and may provide an alternative to the use of iPSCs for the generation of patient- or disease-specific neurons [42,43]. Directly reprogrammed neurons have been shown to retain many important aspects of the donor-derived starting fibroblasts [42,44], with the main advantage that the resulting neurons maintain, at least partially, the age of the donor [45,46].

However, extended 2D culture for mature neurons has several limitations, in that, it fails to mimic (i) complex cell–cell interactions, (ii) endogenous tissue architecture, and (iii) the spatial gradient of the growth factors critical for regional specification of the human brain [47,48]. This precludes the possibility of studying later phases of DA neuron differentiation and modeling the intricacy of functional neural networks, as well as investigating development and disease in a much more physiologically relevant cellular context (Table 1).

### 2.2. Cell Transplantation

The potential capability of transplanted neurons to replenish lost DA neurons has long been considered a promising strategy to treat motor symptoms of PD caused by the focal and selective degeneration of the A9 DA neuron subtype. Pre-clinical studies have shown that the transplanted neurons were able to repair damaged neural circuitry and restore DA neurotransmission by re-forming afferent and efferent connections in lesioned 6-OHDA PD rats. In addition, reports on stem cell treatments for brain repair provide firm evidence as to the feasibility of investigating the mechanisms of hPSC-derived DA neuron differentiation in xenograft animal models [49]. 

Initial proof of concept studies for cell-based therapy in PD were performed in the late 1980s, with the transplantation of fetal VM tissue into PD patients. While these studies confirmed the safety and efficacy of this approach—as demonstrated by clinical improvement, restoration of striatal DA release, and long-term survival of grafted neurons—beneficial effects were only observed in a subset of patients [11,50]. Furthermore, limited tissue availability and ethical issues have so far hindered the widespread use of human fetal tissue in cell replacement therapy, making it hard to scale up and standardize treatment [51,52]. In fact, fetal tissue is not expanded in a dish but is directly transplanted once dissected from the embryo. The outcome is thus impacted by variability in tissue quality and dissection. Recreating human DA neurons from hPSCs in vitro opens up unprecedented opportunities to access a renewable source of cells that are potentially suitable for large-scale PD therapeutic applications. Pioneering research in preclinical animal models of PD demonstrated with a side-by-side comparison that the transplanted fetal- and stem-cell-derived DA neurons are phenotypically similar, and that they survive, innervate, and function with equal potency, after transplantation.

Homotopic graft placement is optimal for studying physiological signaling and appropriate afferent regulation. However, achieving meaningful projections from SNpc to appropriate regions within the striatum, via homotopic grafting in a clinical setting, may be more challenging, due to the much larger size of adult human brain as compared to a rat [53,54]. Therefore, in preclinical studies, the protocol for transplanting DA neurons into a PD model most often involves grafting DA progenitors into the striatum, a heterotopic graft location [19,55,56]. This is also the placement in fetal VM and PSC-based clinical trials [52].

Although A9 cannot be unequivocally distinguished from A10 neurons in 2D in vitro culture, due to the absence of topological and spatial landmark and a lack of definitive markers, the transplantation of these cells at progenitor stage into PD rat models can still represent a valid strategy to recreate human DA heterogeneity. Indeed, after homotopic intranigral xenografting of hPSC-derived DA neurons, these cells display long-term survival and continue to mature and differentiate. Transplanted DA neurons anatomically and functionally integrate in the SNpc of the host rodent brain, projecting over long distances and restoring motor function by re-establishing axonal connectivity, and subsequently, the DA transmission. Several studies provided compelling evidence that both A9- and A10-like neurons are formed in the graft with the capacity for long-distance target-specific innervation [19,57,58].

Besides having an important role in preclinical studies, stem cell transplantation in PD animal models also represents a powerful strategy to achieve heterogeneity of neurons in the midbrain, and may allow studies of the molecular mechanisms underlying DA subtype specification. Novel technologies including chemo- and optogenetic approaches have made the assessment of synaptic integration of hPSC-derived neurons more accessible, laying the groundwork for a more detailed analysis of the transplant function, and offering the possibility of linking the profile of these neuronal subpopulations with their function and innervation targets. Recent studies used graft-initiated monosynaptic retrograde rabies virus tracing, to track the extent of graft-to-host connectivity, showing that the transplanted neurons establish appropriate afferent synaptic connections with host neurons, thus, highlighting their functional integration into host circuitry [57,59,60,61,62]. Grafted DA neurons project to target regions normally innervated by either A9 neurons located in SNpc or A10 neurons in VTA [55]. All these findings provide evidence that recreating axonal guidance and targeting recognition mechanisms in PD-lesioned denervated rat forebrain can drive growing axons to their appropriate target regions, and that these mechanisms are sufficiently specific and refined to distinguish between A9 and A10 subtypes. 

One of the main advantages of stem cell transplantation lies in the fact that it exploits the host environment to maintain DA xenografts in the long term, mimicking both DA neuron circuits and the complexity of DA neuron development (Table 1). This may lead to the development of more precise A9-specific differentiation protocols that are required for successful intranigral grafting and a more complete functional circuitry repair for clinical use. In contrast, the re-isolation of fresh human grafts in the rodent brain and subsequent single-cell dissociation for transcriptomics and proteomics analysis is extremely challenging and may represent a considerable hurdle. This is because re-isolation damages the most sensitive and functionally mature DA neurons, thereby hampering any in-depth molecular analysis and the acquisition of a comprehensive dataset of the human DA neuronal spectrum (Table 1).

### 2.3. 3D Culture

Human brain organoids represent a major emerging advancement in stem-cell technologies, as they allow hPSCs to be arranged into a 3D organ-like structure, providing an extraordinary opportunity to explore human brain development in vitro in a physiological cell context (Figure 2A) [63,64]. These organoids rely on the remarkable properties of hPSCs to endogenously produce morphogenetic and patterning factors that guide self-organization and self-patterning in multiple regions of the human brain. Indeed, initial studies performed in the absence of extrinsic growth factors reported that differentiation of hPSCs in brain organoids recapitulated various regions of the human brain, as well as fundamental aspects of development. Given their heterogeneous cellular composition, these 3D structures were also called cerebral organoids, as they contain multiple regional identities resembling distinct areas of the human brain. During the initial phase of cerebral organoid differentiation, neural progenitors radially organize into rosette cell structures that give rise to an apicobasally polarized neuroepithelium, which further develops into cortical-like multilayers displaying features of human corticogenesis. These organoids generate a broad diversity of cell types including neuroepithelial progenitors, cells from the cerebral cortex and the retina, and glial cells, as well as contaminants from mesoendoderm germ layers [65,66,67].

Currently, 3D human brain organoids can be generated following several different protocols based on the addition of regionalizing factors. These direct the emergence of distinct cytoarchitectural features, thereby recreating a specific brain-tissue type. In order to efficiently obtain higher-order brain functions in a dish, all human cell-based 3D model systems necessarily require a scaffold supporting cellular self-organization into more complex architectures within a 3D geometric space. Matrigel, a protein mixture resembling the extracellular matrix, is commonly used to sustain the structure and induce the correct neuroepithelial apical-basal polarity in floating culture conditions. 3D culture systems are also enhanced by bioreactors that enable the diffusion of nutrients and oxygen, as well as the absorption of patterning factors. These steps allow long-term culture of brain organoids, leading to the generation of mature neuronal features with electrical properties and the formation of an intricate neuronal circuitry map. Long-term culturing also increases the heterogeneity of cell types over time. It promotes the development of glial cells, which play a critical role in cell–cell interactions and circuit connectivity, and preserve neuronal homeostasis (Table 1) [68,69,70,71].

Recent studies report that hPSCs can be patterned toward a midbrain fate, resulting in regionalized human VM organoids that exhibited postmitotic molecular features and electrophysiological activity. Interestingly, morphological analysis of these VM organoids revealed the presence of intracellular and extracellular neuromelanin, visible as dark granular pigmentation after long-term culture, resembling that released in SNpc in primate VM. These findings indicate that functionally mature DA neurons emerge within VM organoids and diversify into different subtypes, including a pigmented A9-like DA population [21,72,73]. 

However, if on one hand VM organoids offer a unique opportunity to study molecular features of DA neuron subtypes, questions relating to how distinct DA identities are linked to functions and axonal innervation targets remain unresolved. 

Recent studies modeled dorso-ventral patterning of mouse and human embryonic stem cells, using microfluidic diffusion-based gradient generators. This technology recreates morphogen gradients during brain development, potentially identifying new markers and providing greater insights into the underlying mechanisms of human brain development [74,75]. However, obtaining mature cells and mimicking the intricacy and cell–cell interaction of different brain regions and in these systems, remains challenging. These issues could be addressed by using a next-generation 3D culture approach in which brain organoids of different regional identities are fused in a dish. So-called “assembloids” are able to recreate cell–cell interactions and neuronal migration and reconstruct axonal projection patterns between different brain regions (Figure 2B) [76,77]. These systems are already used to model cortical interneuron migration by assembling ventral and dorsal forebrain organoids. However, no compelling evidence has as yet indicated that the assembloid platform may be useful in recreating the later stages of the DA developmental program and the molecular cues governing appropriate DA axonal targeting, in a system that is more tractable than xenografts (Table 1). 

Given the resemblance of brain organoid models to early stages of cortical neurogenesis, they have also been used to model neurodevelopmental disorders, in an attempt to find potential treatments. By using patient-derived iPSCs carrying genetic mutations, conditions such as microcephalies have been modeled in brain organoids, identifying disease pathways underlying these neurological disorders [78,79,80]. VM organoids derived from differentiated PD iPSCs and isogenic control lines were used to model pathological phenotypes by recapitulating the hallmarks observed in PD patients [72]. However, the fact that PD is the product of a complex interaction of both genetic and environmental factors has made the modeling of this disease in brain organoids extremely challenging. Efforts have been made to recreate the pathological environment of PD in a dish by omitting the use of brain-derived neurotrophic factor (BDNF) and antioxidants, and adding various stressors such as rotenone and a mixture of human alpha-synuclein preformed fibrils in culture. This would trigger progressive DA neuron cell death, extensive proteinopathy, and inflammation, mimicking the milieu in the diseased patient brain. iPSCs derived from PD patients carrying the *LRRK2*-G2019S mutation were differentiated in VM organoids, with the aim of studying the mechanisms of degeneration. *LRRK2*-G2019S, a missense mutation in the leucine-rich repeat kinase 2 (*LRRK2*) gene locus, is primarily associated with alpha-synuclein accumulation and mitochondrial dysfunction and represents the most commonly known cause of late-onset familial and sporadic PD. Several reports showed that disease-relevant phenotypes in PD patients, including increased aggregation of alpha-synuclein and its aberrant clearance, were recapitulated in 3D culture, closely mimicking those seen in patients with mutant *LRRK2*-associated sporadic PD (Figure 2C) [73,81,82].

However, the remarkable potential of brain organoids to model later developmental stages and generate more mature neurons has to date been hampered by their poor reproducibility and incomplete maturation. Moreover, as organoids increase in size, access to oxygen and nutrients in the inner layers is progressively limited, thereby affecting organoid function and lifespan (Table 1). Recent studies describe two very different strategies to generate homogenous, mature, and healthy organoids. One is based on the creation of a 2D scaffold using individual inert fibers to guide the self-organization of hPSCs into a 3D structure with increased reproducibility, while the other exploits the ectopic expression of genes to induce a vascular-like structure within the organoid, thereby decreasing the volume of the necrotic core and enhancing functionality [83,84,85,86,87]. Major efforts are aimed at generating more favorable growth and differentiation conditions facilitating the delivery of oxygen and patterning factors, in order to better recapitulate the physiologically relevant aspects of developing human brain tissue and to extend the use of organoids in disease modeling and drug discovery [85]. Incorporating microfluidic channels in human brain organoids brings several attractive benefits. Most importantly, microfluidic technology allows the delivery of oxygen and nutrients and removal of waste products in 3D tissue structures [88,89]. Furthermore, brain organoids with perfused channels are engineered in microfluidic devices by exploiting the effects of capillary forces and laminar flow. They are now making an important contribution to dissecting complex phenomena including paracrine signaling and cellular interaction and migration.

## 3. Single Cell Sequencing in Decoding Human Brain Complexity

Major advances in next-generation sequencing technologies mean that it is now possible to characterize the genome, epigenome, and transcriptome at single-cell resolution [13,90]. Single-cell RNA sequencing (scRNAseq) has emerged as a powerful methodology that is able to profile the full transcriptome of individual cells, providing an unprecedented measurement of heterogeneity, an intrinsic feature of all living biological systems [91]. scRNAseq is able to deconstruct the complexity of fundamental biological processes, by characterizing heterogeneous and rare cell populations and identifying novel cell types and regulatory gene networks, thereby elucidating the molecular mechanisms controlling human brain functions. For the analysis of brain tissues, cell dissociation is the most critical step, as conditions in this phase directly affect the molecular profiles of cells. In neuroscience, single nuclei RNAseq is rapidly replacing scRNAseq, since neurons are hard to capture using single-cell dissociation protocols [92,93]. Several technologies for library preparation based on microdroplet technology are reported, such as 10X genomics and Drop-Seq [94]. In these methods, a cell/nucleus, reaction enzymes, and a barcoded bead are encapsulated in an oil droplet. Reverse transcription, including tagging each RNA with a cell-specific barcode takes place within each droplet. As an alternative to droplet-based technologies, microwell protocols, including Smart-Seq, can be deployed. Although these protocols generally have lower throughput, they are able to capture full-length transcripts and have a better sensitivity for detecting lowly expressed genes [95,96]. More recently, combinatorial single cell indexing platforms, based on split-and-pool DNA-barcoding strategies, offer another highly scalable approach comparable to the droplet systems, but without the need for specific instruments. The large number of cells that can now be sequenced combined with the relative sparsity of data, has posed unique data science challenges, including handling of data sparsity, robust differential expression, and accurate trajectory inference.

For years, scientists have been intrigued by the fact that DA neurons, which make up less than 1% of the total number of brain neurons, can play such diverse roles in influencing a large spectrum of cognitive and motor functions. With the advent of single-cell technologies, research efforts have aimed to create a systematic catalog of molecularly distinct DA neuron subtypes exhibiting specific physiological and functional properties, in order to understand their different vulnerability in PD and to develop more effective pharmacological and cell-based treatments [15,49,91]. 

Recent studies examined DA neuron subtypes in mouse and humans, based on their molecular profiles at single-cell resolution. In transgenic adult mice, isolation based on Pitx3 expression and the subsequent sequencing of single neurons, identified five molecularly distinct DA clusters in the midbrain. It exhibited the different gene expression levels of dopamine-producing machinery components, suggesting that each one has a distinct role in mammalian VM [97]. Similarly, a single-cell dataset comparison between fetal VM dissected from human embryos (6–11 weeks) and mouse, captured the similarities and differences in VM development between the two species [26]. The findings of this investigation revealed the evolutionary conservation of cell type diversity in humans and mice, highlighting the major differences in proliferation and timing of DA development. However, despite recent efforts, classifying neurons into subtypes remains far from straightforward. All the subgroups are closely related, as they all display a typical DA neuron signature expressing the genes required for DA synthesis but possess unique molecular features with a small set of differently expressed genes. To identify rare subtypes, a large number of cells need to be captured in order to ensure sufficient representation, especially considering that DA neurons make up only a tiny fraction of the cells captured when sequencing the mammalian midbrain. At the same time, techniques with high throughput (e.g., 10×, sci-fi-seq) have lower sensitivity than, for example, Smart-seq, meaning that a delicate decision has to be taken on whether to focus on the number of analyzed cells versus sensitivity. Another complicating factor is that the transcriptome alone might not be sufficient to define closely related DA subtypes. This limit can be overcome by using multi-omics approaches, discussed in detail in the following subsection.

## 4. Advances in Single Cell Transcriptomics

scRNAseq is now being combined with morphological characterization, lineage tracing, and functional analysis, in order to compile a comprehensive catalog of molecularly distinct neuronal subtypes, a valuable asset for elucidating their development and function in human brain [98].

### 4.1. Spatial Transcriptomics

Spatial transcriptomics is used to better understand the function of a single cell in distinct spatial architectures within a brain tissue sample, as well as to determine the subcellular localization of RNAs (Figure 3A) [99,100]. Spatial transcriptomics is a revolutionary technology and is based on the assumption that in order to define the function of specific cell types, it is necessary to obtain positional information while preserving cell–cell communication and intracellular regulation. A tissue section is placed on an array containing probes, to capture gene expression data. This results in a topologically-based transcriptional deconvolution of human tissue, which is required to provide a molecular identity map illustrating how cell types are localized within the tissue. Although spatial transcriptomics does not require cell dissociation, thus, providing a better understanding of the role of an individual cell within its microenvironment, sequencing resolution is significantly lower than in conventional single-cell approaches and is also highly dependent on the structure of tissues and the harvesting procedures [101].

### 4.2. Cellular Barcoding

Lineage tracing tracks the diversification history of cellular subtypes generated during development. This technology labels an individual cell by introducing heritable genetic makers called “barcodes” at early time-points, in order to trace the state of its clonal progeny at different stages of stem cell differentiation, via sequencing (Figure 3B) [102]. Barcoding is a lentiviral cell tag method usually consisting of a 5–7 bp fixed region and a 10–12 bp variable region, which represent a unique identifier for individual cells. A combination of 4–5 tags can be used to increase barcoding library tracking. By harvesting cells at different stages, single cell computational inference analysis is able to place individual cells along a temporal axis, reconstructing a developmental continuum of stem cell states from cell origin to differentiated subtypes, including intermediate progenitors [103]. To reconstruct a composite phylogenetic tree that accurately defines branch points and sister cells from each division, single-cell chromatin accessibility assays such single-cell ATAC-seq can be adopted to add genomic and epigenomic details of these heritable properties. Reconstructing lineage history and mapping progeny relationships deduces cell fate decisions toward specific neuronal subtypes, and provides valuable insights into how to best direct and refine stem cell differentiation for use in biomedical applications [104]. By employing machine learning and gene regulatory network analysis, it is becoming increasingly possible to identify new regulators of differentiation that trigger a cascade of transcriptional events, leading to cell populations with a functionally mature phenotype.

### 4.3. Multimodal Single Cell Data

Multimodal single-cell data transcriptomics alone is often unable to separate molecularly similar, but functionally distinct, neuronal cells. Multimodal single-cell technologies, which simultaneously profile multiple data types in the same cell represent the latest frontier in the discovery and characterization of cell states [105,106]. For example, utilizing oligonucleotide-tagged antibodies, in conjunction with scRNAseq, a method called CITE-seq, was used to simultaneously quantify RNA and surface protein abundance in single cells, via the sequencing of antibody-derived tags [107,108]. Recent advancements now enable the simultaneous profiling of transcriptome, alongside either chromatin accessibility, methylation, or nucleosome occupancy. Combined analysis of single-cell transcriptomes and proteomics has proven more challenging, as these are vastly different molecular modalities. Converting them into a uniform state that can be analyzed together in an “omics-wide” manner is a substantial technical challenge.

### 4.4. Patch-Seq

Patch-seq is a specific example of multimodal single-cell sequencing, combining electrophysiological whole-cell patch-clamp recordings, scRNAseq, and morphological characterization, in order to obtain the functional classification of neuronal subtypes (Figure 3C) [109]. Of note, single-cell Patch-seq is able to correlate the molecular features of diverse neuronal types with their physiological and functional role in the brain. After electrophysiological recordings, the cell contents are aspirated through the patch-clamp pipette and are prepared for scRNAseq. This technology represents an important step forward in providing high-resolution analysis of rare cell types or a specific subset of neurons, which would otherwise be lost among the thousands of single cells, when using conventional scRNAseq approaches [110,111]. Single-cell analysis of electrically dysfunctional neurons is also extremely important for studying the etiology of neurological disorders. Healthy and diseased iPSCs have been widely used to model brain diseases, in order to explore aberrant phenotypes of specific neuronal populations, using single cell Patch-seq.

## 5. Conclusions

Significant advances have been made in the field of stem cell biology in establishing differentiation protocols driving hPSCs toward specific neuronal cell types that exhibit distinct physiological and functional properties. The growing number of human stem cell models used to build human tissue and organs, together with advanced single-cell sequencing technologies, is allowing us to peek into the complexities of the human brain.

The most commonly used method to characterize the phenotype and molecular features of progenitors and neuronal subtypes in culture is to generate transcriptional and epigenetic profiles and compare them to their in vivo counterparts. However, the inability to access the human brain and the scarcity of human post-mortem samples make such an approach extremely difficult, if not almost impossible. Attempts to recapitulate the main features of the human brain in a laboratory setting are at risk of remaining fruitless. These difficulties have propelled different labs across the world to share a common mission to generate a comprehensive and detailed human brain atlas—a dataset based on adult and fetal human samples. This would serve as the gold standard for analyzing the quality of in vitro-generated neurons and determining the authenticity of their acquired cell fate. 

A side-by-side comparison with fetal VM cells has already provided great insights into how hPSCs can be manipulated into becoming functionally mature DA neurons and has led to the development of differentiation protocols used for cell-based therapy in PD. However, there is still a long way to go before we are able to obtain a complete molecularly defined map of human DA neuron diversity. To develop human cell-based models that more faithfully resemble mature tissues, it would be highly advantageous to exploit the precious resource of human fetal cells. This would establish new and advanced primary culture systems or in vivo xenografts aimed at driving human fetal cells, to differentiate further into subtype-specific maturation. Building mature human tissue from its fetal counterpart would allow the recapitulation of the key authentic molecular aspects of late human DA neurogenesis, which, coupled with cutting-edge next-generation sequencing technologies, may define molecularly distinct identities of mature human DA neurons at the single-cell resolution, reconstructing the positional and functional properties and fate trajectories of developing human VM tissue. Future efforts in these directions will be crucial in dissecting the heterogeneity behind DA neuron biology and in studying the physiological and pathological mechanisms underlying the selective vulnerability of A9 neurons in PD.

## Figures and Tables

**Figure 1 cells-10-01366-f001:**
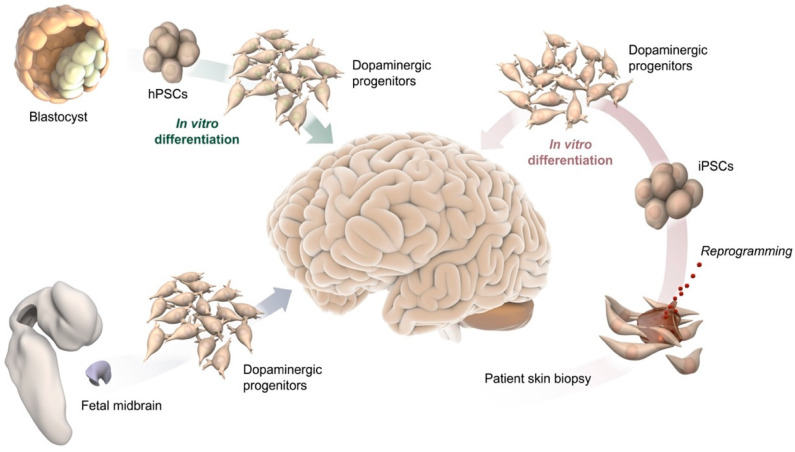
Schematic illustration of potential DA progenitor sources for modeling VM differentiation and for therapeutic application in PD, including blastocysts and their pluripotent stem cell derivative (top-left), fetal midbrain tissue (bottom left), and somatic cells from adult individuals, such as a skin biopsy used to reprogram cells to pluripotency (bottom-right).

**Figure 2 cells-10-01366-f002:**
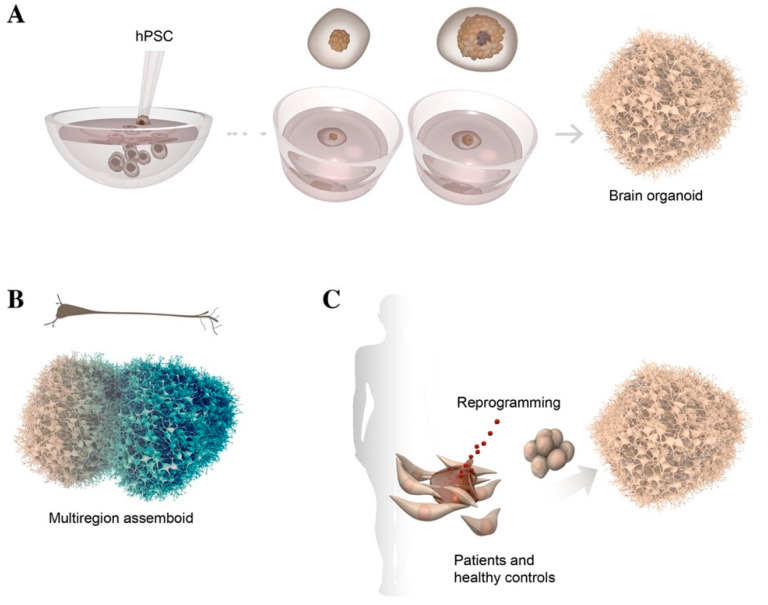
Schematic illustration of hPSC differentiation into human brain organoid using conic wells and Matrigel embedding (**A**). Representation of multiregion assembloid formed by the fusion of two distinct regionalized brain organoids; neurons mature and project over long distances, reaching their target structures (**B**). Somatic cells from healthy and PD patients reprogrammed into induced pluripotent stem cells and differentiated in brain organoid (**C**).

**Figure 3 cells-10-01366-f003:**
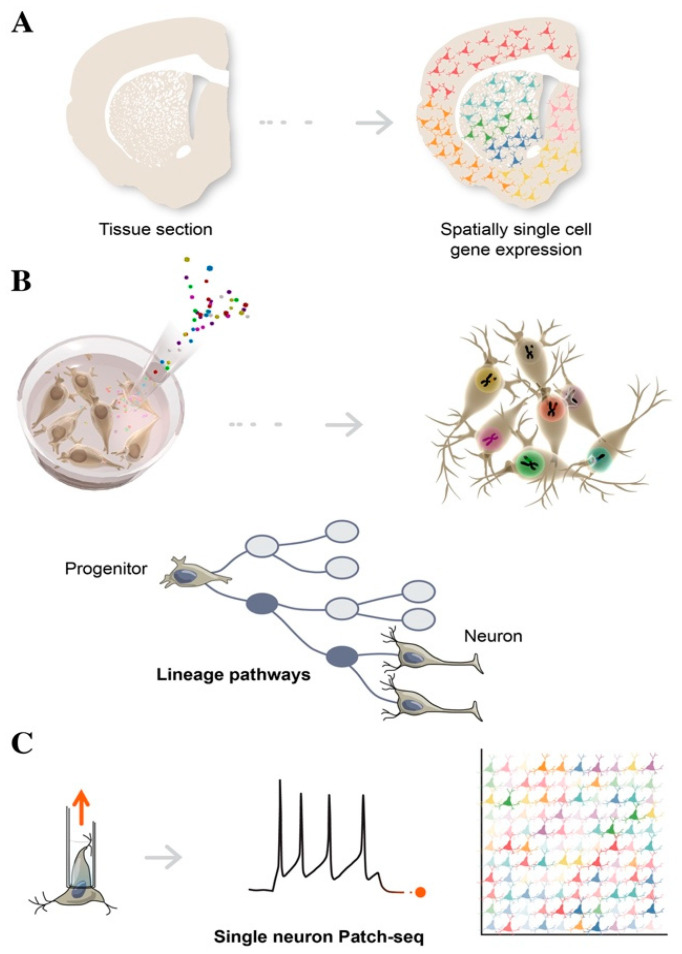
Schematic illustration of single-cell technologies combining transcriptomics with morphological features, to assign cell types to their locations in histological sections (**A**), with developmental lineage tracing to track cellular relationships during development or regeneration (**B**), and with functional analysis to simultaneously profile single-cell transcriptome and electrophysiological properties (**C**).

**Table 1 cells-10-01366-t001:** Comparison of human stem cell models.

	*2D* *Cell Culture*	*3D* *Cell Culture*	*Assembloid Culture*	*Cell Transplantation*
Ease of experimental set-up	*✓*	*✓*	*✓*	*✓*
Generation of functionally mature neurons	*✓*	*✓*	*✓*	*✓*
Reproducibility	*✓*	*✓*	*✓*	*✓*
Genetic manipulation	*✓*	*✓*	*✓*	*✓*
Single cell dissociation	*✓*	*✓*	*✓*	*✓*
Cell diversity	*✓*	*✓*	*✓*	*✓*
Cell-cell interaction	*✓*	*✓*	*✓*	*✓*
Tissue viability	*✓*	*✓*	*✓*	*✓*
Amendable for HTS	*✓*	*✓*	*✓*	
Fully humanized	*✓*	*✓*	*✓*	

*✓**Optimal*; *✓**Good*; *✓**Feasible*.

## Data Availability

Not applicable.

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
