# Peer review of "Dopamine Neuron Diversity: Recent Advances and Current Challenges in Human Stem Cell Models and Single Cell Sequencing"

_cells, 2021, doi:10.3390/cells10061366_

Round 1
Reviewer 1 Report
In this Review article, Fiorenzano et al. discuss different in vitro and in vivo human stem cell-based models for the study of midbrain dopaminergic neuron development and disease, particularly in regard of the meanwhile well-known molecular and functional heterogeneity of these neurons in the adult human brain. Furthermore, the authors discuss recent advances in single-cell transcriptomic analyses as a versatile tool to deconstruct the molecular heterogeneity of human midbrain dopaminergic neurons.
The Review article by Fiorenzano et al. already provides an excellent overview on human stem cell-based in vitro and in vivo models for midbrain dopaminergic neuron generation and recent advances in single-cell transcriptomics to study the broad diversity of these neurons. However, some points still require the attention of the authors:
- Although the authors already address the major advantages and disadvantages of each human stem cell-based model (in vitro and in vivo) at the end of the corresponding sections (2.1. to 2.3.), this Review article would profit enormously from a sort of “stratification” of the usefulness and applicability of the corresponding model at different levels: molecular, cellular, tissue, organ and systemic (whole body including environmental interactions). Each model probably outperforms the others at individual levels, and researchers should become aware that it depends very much on the questions they are asking to choose the right model. This could also be achieved by a summarizing table or a more informative illustration than the current Figure 2. In Figure 2, it is hard to understand what the two cells (and corresponding organoid cultures) in the middle of the top panel A should represent, and the same is true for panel C in this figure. For what are these organoids or assembloids useful, or what should be done with them?
- Given the current ethical debate about stem cell-derived early human embryos and human-primate embryo chimaeras, the term “in vivo human stem cell-based models” and the sentence in line 76 (“The generation of human tissues in vitro and in vivo …”) may be misunderstood and misleading. In a strict sense, transplantation of human cells or neurons into the brains of other species represent a heterologous model and do not correspond to a genuine human “in vivo” model, as species-specific differences in the environment of the transplant may also have to be considered in this case. Moreover, cell transplantation models are necessarily coupled to a 2D (and 3D?) culture model, and thus question the current listing as a separate model by the authors. Perhaps a stratification according to the intended use or application of the corresponding model (see comment 1. of this reviewer) would help to streamline this part of their Review article.
- In section 2.2. “Cell transplantation” on pages 4 and 5, the authors mention exclusively intranigral grafting experiments and write (lines 200-202): “One of the main advantages of stem cell transplantation lies in the fact that it exploits the host environment to maintain DA xenografts in the long term, mimicking both DA neuron circuits and the complexity of DA neuron development.” However, most of the transplantation approaches, even in the human brain, have grafted the stem cell-derived dopaminergic cells into the striatum. The striatum is not the site where these cells usually reside and can thus be presumed to provide other or very different environmental cues than the rodent or human ventral midbrain. This may be a very important point to consider, particularly in the context of studying human DA neuron development, and it will be interesting to know the authors’ opinion in this regard.
- In section 2.3. (2.2. in the manuscript) “3D culture”, the authors explain the advantages and disadvantages of the classical organoid technology, and refer to assembloids and vascularized organoids (to improve oxygen and nutrient exchange) as newer developments in this field. However, they may also discuss the usefulness of microfluidic “scaffolding or patterning” approaches, as recently reported by Rifes et al., 2020 (Nat. Biotechnol. 38 (11), 1265-1273)), in this context.
- Section 3.1. (or just 3.?) “Single cell sequencing in decoding human brain complexity”: At the very end of this section, the authors discuss the fact that single-cell sequencing of dopaminergic neurons in the mouse and human ventral midbrain to classify these neurons into individual subtypes is not a “straightforward” technique, because “All the subgroups are closely related as they all display a typical DA neuron signature expressing genes required for DA synthesis but possess unique molecular features with a small set of different expressed genes”. It will be interesting to know the authors’ opinion about possible bias that may be introduced by the nature of such large-scale data analyses.
- In section 4. “Conclusions”, the authors may briefly comment the usefulness of human stem cell-based models for the study of age-dependent neurodegenerative diseases in the light of recent data showing that direct conversion (or “reprogramming”) of patient-derived somatic cells (e.g., adult fibroblasts) to so-called induced neurons may be the better choice to study disease mechanisms and eventually therapies in these cases. See also the fourth issue in the next comment of this Reviewer.
- Some editing of the text or clarification of the intended meaning are required in the following instances:
- Page 1, line 17: “Although many years of scientific efforts in mouse and adult brain have positioned us …”. The authors should clarify what they mean with “in mouse and adult brain” – mouse embryonic and adult brain, human brain?
- Page 1, lines 43 and 44: “A8 and A10 neurons innervate the nucleus accumbens, septum, and prefrontal cortex in the ventral striatum via the mesolimbic pathway …”. “Prefrontal cortex in the ventral striatum” is wrong and misleading.
- Page 2, lines 49 and 50: “It is A9 neurons that are primarily degenerated in PD and are thus more vulnerable to oxidative damage, …”. It is not clear why A9 neurons should be more vulnerable to oxidative damage if they degenerate primarily in PD?
- Citations on page 3, lines 99 to 103: “The study of human CNS diseases in animal model systems has therefore proved inadequate [26]. Investigations into neurodegenerative diseases such as PD and Alzheimer’s using animal modeling are even more problematic, since the etiology of these disorders is usually complex and diverse and is further complicated by individual risk variants, distinct genetic and epigenetic backgrounds, and unknown environmental triggers [27, 28].” The authors cite the work by La Manno et al., 2016 (direct comparison of single-cell RNA-Seq results of the developing midbrain dopaminergic domain in the mouse and human ventral midbrain at different embryonic stages, and of in vitro differentiated human pluripotent stem cells, citation 26), which has not proven “the study of human CNS diseases in animal model systems” as inadequate. They also cite the work by Caiazzo et al., 2011 (reporting the direct conversion of mouse and human fibroblasts to induced dopaminergic neurons, citation 27) and Drouin-Ouellet et al., 2017 (addressing the fact that direct conversion of adult human fibroblasts is much more inefficient than reprogramming of young or fetal fibroblasts and identified REST as one of the key mediators of this effect, citation 28) in this context, which again are not related to the statements made in the text. The authors should provide more adequate citations in this context.
- Page 4, lines 132 and 133. “Interestingly, the authors showed that DA neurons are not derived from neuroepithelial cells but from VM floor plate progenitors …”. Are VM floor plate cells not neuroepithelial cells, i.e., are they not part of the neuroepithelium?
- Page 4, lines 136 to 138: “Dual SMAD inhibitors blocking TGFb signaling, combined with the ventralizing and caudalizing factors SHH and GSK3, …”. GSK3 itself is not used in the differentiation protocols, but a GSK3 inhibitor for WNT pathway activation. Moreover, the dual SMAD inhibitors target both the TGFb and BMP pathways. This should be corrected.
- Page 4, line 166: “lesioned 6-OHD PD rats”. It should be 6-OHDA.
Author Response
In this Review article, Fiorenzano et al. discuss different in vitro and in vivo human stem cell-based models for the study of midbrain dopaminergic neuron development and disease, particularly in regard of the meanwhile well-known molecular and functional heterogeneity of these neurons in the adult human brain. Furthermore, the authors discuss recent advances in single-cell transcriptomic analyses as a versatile tool to deconstruct the molecular heterogeneity of human midbrain dopaminergic neurons.
The Review article by Fiorenzano et al. already provides an excellent overview on human stem cell-based in vitro and in vivo models for midbrain dopaminergic neuron generation and recent advances in single-cell transcriptomics to study the broad diversity of these neurons.
We thank the Reviewer for their overall positive comments on the work presented. Their careful and insightful remarks and suggestions have enabled us to considerably improve the quality of the manuscript. Please find below a point-by-point response to all comments.
However, some points still require the attention of the authors:
- Although the authors already address the major advantages and disadvantages of each human stem cell-based model (in vitro and in vivo) at the end of the corresponding sections (2.1. to 2.3.), this Review article would profit enormously from a sort of “stratification” of the usefulness and applicability of the corresponding model at different levels: molecular, cellular, tissue, organ and systemic (whole body including environmental interactions). Each model probably outperforms the others at individual levels, and researchers should become aware that it depends very much on the questions they are asking to choose the right model. This could also be achieved by a summarizing table or a more informative illustration than the current Figure 2. In Figure 2, it is hard to understand what the two cells (and corresponding organoid cultures) in the middle of the top panel A should represent, and the same is true for panel C in this figure. For what are these organoids or assembloids useful, or what should be done with them?
We are grateful to the Reviewer for suggesting that we include a table, which will undoubtedly provide a clear summary for the readers. We have now included a table on page 8 summarizing the advantages and disadvantages of different aspects of human stem cell models that we have discussed in this manuscript. The aim of the table is also to illustrate to the reader that there is no one "best" human stem cell system for modeling development and disease, but that the choice depends very much on the underlying biological questions being asked.
- Given the current ethical debate about stem cell-derived early human embryos and human-primate embryo chimaeras, the term “in vivo human stem cell-based models” and the sentence in line 76 (“The generation of human tissues in vitro and in vivo …”) may be misunderstood and misleading. In a strict sense, transplantation of human cells or neurons into the brains of other species represent a heterologous model and do not correspond to a genuine human “in vivo” model, as species-specific differences in the environment of the transplant may also have to be considered in this case.
We fully agree with the Reviewer and have modified the heading of section 2 accordingly (on page 2 line 87).
Moreover, cell transplantation models are necessarily coupled to a 2D (and 3D?) culture model, and thus question the current listing as a separate model by the authors. Perhaps a stratification according to the intended use or application of the corresponding model (see comment 1. of this reviewer) would help to streamline this part of their Review article.
We agree with the Reviewer that transplantation is usually coupled with 2D culture. However, we have decided to maintain the original structure of the section in order to include fetal tissue, which is directly transplanted once dissected from the human embryo.
- In section 2.2. “Cell transplantation” on pages 4 and 5, the authors mention exclusively intranigral grafting experiments and write (lines 200-202): “One of the main advantages of stem cell transplantation lies in the fact that it exploits the host environment to maintain DA xenografts in the long term, mimicking both DA neuron circuits and the complexity of DA neuron development.” However, most of the transplantation approaches, even in the human brain, have grafted the stem cell-derived dopaminergic cells into the striatum. The striatum is not the site where these cells usually reside and can thus be presumed to provide other or very different environmental cues than the rodent or human ventral midbrain. This may be a very important point to consider, particularly in the context of studying human DA neuron development, and it will be interesting to know the authors’ opinion in this regard.
This is an important point, and we are grateful to the Reviewer for bringing it up. We now discuss this issue in the revised version of the manuscript on page 5 line 189.
We agree with the Reviewer that most transplantation approaches in PD models described in the literature are based on heterotopic graft placement of DA progenitor transplantation in the striatum in PD animal models. This is mainly dictated by clinical reasons as most of these studies aimed to test the efficacy of stem cell-derived neurons in releasing dopamine and restoring motor functions in order to improve cell-based therapy for PD.
In fact, although homotopic transplantation in SNpc exploits a more physiological context in which to develop functionally mature DA neurons, it is also very challenging to obtain full and robust axonal innervations from homotopic grafts towards A9 and A10 target brain regions. This is even more difficult in human brain, where SNpc is much further from the striatum than in rodents, meaning that DA neurons need to project over very long distances before releasing dopamine and restoring motor functions. Based also on these practical considerations, in ongoing clinical trials the stem cell product is transplanted directly into the striatum in PD patients.
- In section 2.3. (2.2. in the manuscript) “3D culture”, the authors explain the advantages and disadvantages of the classical organoid technology, and refer to assembloids and vascularized organoids (to improve oxygen and nutrient exchange) as newer developments in this field. However, they may also discuss the usefulness of microfluidic “scaffolding or patterning” approaches, as recently reported by Rifes et al., 2020 (Nat. Biotechnol. 38 (11), 1265-1273)), in this context.
We thank the Reviewer for raising this point. We now discuss microfluidic approaches in the revised version of the manuscript on page 6 line 282 and on page 8 line 338.
Section 3.1. (or just 3.?) “Single cell sequencing in decoding human brain complexity”: At the very end of this section, the authors discuss the fact that single-cell sequencing of dopaminergic neurons in the mouse and human ventral midbrain to classify these neurons into individual subtypes is not a “straightforward” technique, because “All the subgroups are closely related as they all display a typical DA neuron signature expressing genes required for DA synthesis but possess unique molecular features with a small set of different expressed genes”. It will be interesting to know the authors’ opinion about possible bias that may be introduced by the nature of such large-scale data analyses.
We thank the Reviewer for suggesting that we include some possible challenges on this aspect in the revised version of the manuscript on page 9 line 393.
- In section 4. “Conclusions”, the authors may briefly comment the usefulness of human stem cell-based models for the study of age-dependent neurodegenerative diseases in the light of recent data showing that direct conversion (or “reprogramming”) of patient-derived somatic cells (e.g., adult fibroblasts) to so-called induced neurons may be the better choice to study disease mechanisms and eventually therapies in these cases. See also the fourth issue in the next comment of this Reviewer.
We thank the Reviewer for their suggestion. We now discuss direct reprogramming in the 2.1 section on page 4 line 146.
- Some editing of the text or clarification of the intended meaning are required in the following instances:
- Page 1, line 17: “Although many years of scientific efforts in mouse and adult brain have positioned us …”. The authors should clarify what they mean with “in mouse and adult brain” – mouse embryonic and adult brain, human brain?
We have rephrased the sentence according to the Reviewer’s suggestion.
- Page 1, lines 43 and 44: “A8 and A10 neurons innervate the nucleus accumbens, septum, and prefrontal cortex in the ventral striatum via the mesolimbic pathway …”. “Prefrontal cortex in the ventral striatum” is wrong and misleading.
We agree with the Reviewer and have changed the sentence accordingly.
- Page 4, lines 132 and 133. “Interestingly, the authors showed that DA neurons are not derived from neuroepithelial cells but from VM floor plate progenitors …”. Are VM floor plate cells not neuroepithelial cells, i.e., are they not part of the neuroepithelium?
We agree with the Reviewer and have now clarified this point.
- Page 2, lines 49 and 50: “It is A9 neurons that are primarily degenerated in PD and are thus more vulnerable to oxidative damage, …”. It is not clear why A9 neurons should be more vulnerable to oxidative damage if they degenerate primarily in PD?
We have rephrased the sentence according to the Reviewer’s suggestion.
- Citations on page 3, lines 99 to 103: “The study of human CNS diseases in animal model systems has therefore proved inadequate [26]. Investigations into neurodegenerative diseases such as PD and Alzheimer’s using animal modeling are even more problematic, since the etiology of these disorders is usually complex and diverse and is further complicated by individual risk variants, distinct genetic and epigenetic backgrounds, and unknown environmental triggers [27, 28].” The authors cite the work by La Manno et al., 2016 (direct comparison of single-cell RNA-Seq results of the developing midbrain dopaminergic domain in the mouse and human ventral midbrain at different embryonic stages, and of in vitro differentiated human pluripotent stem cells, citation 26), which has not proven “the study of human CNS diseases in animal model systems” as inadequate. They also cite the work by Caiazzo et al., 2011 (reporting the direct conversion of mouse and human fibroblasts to induced dopaminergic neurons, citation 27) and Drouin-Ouellet et al., 2017 (addressing the fact that direct conversion of adult human fibroblasts is much more inefficient than reprogramming of young or fetal fibroblasts and identified REST as one of the key mediators of this effect, citation 28) in this context, which again are not related to the statements made in the text. The authors should provide more adequate citations in this context.
We have changed all the references in this part of the manuscript to reflect the statements made in the text.
- Page 4, lines 136 to 138: “Dual SMAD inhibitors blocking TGFb signaling, combined with the ventralizing and caudalizing factors SHH and GSK3, …”. GSK3 itself is not used in the differentiation protocols, but a GSK3 inhibitor for WNT pathway activation. Moreover, the dual SMAD inhibitors target both the TGFb and BMP pathways. This should be corrected.
We have rephrased the sentence according to the Reviewer’s suggestion.
- Page 4, line 166: “lesioned 6-OHD PD rats”. It should be 6-OHDA.
We agree with the Reviewer and have changed the abbreviation.
Reviewer 2 Report
The paper “Dopamine neuron diversity: recent advances and current challenges in human stem cell models and single cell sequencing” by Fiorenzano et al., shows an overview on the advances in the field of dopamine neuron modeling and genomics. The review is overall well organized and most of the appropriate works are covered. Some improvements are still needed in order to be considered for publication as pointed out below:
1) The text needs some English language editing. Several sentences are too long and some spelling mistakes need to be fixed (i.e.” treat human brain disease(s)”)
2) Some references seem to be not consistent with the text where they are cited (i.e.Ref 27,28; 31-33)?
3) The authors should cite some seminal work of Lorenz Studer group on dual/smad inhibition and ESC-derived dopamine neurons (i.e. Chambers et al., 2009; Kriks et al., 2011)
4) The authors should clarify that GSK3 inhibition is used in order to activate WNT pathway that contributes to dopamine neuron specification.
5) The authors state: “Although the absence of topological landmarks in in vitro prevents the precise identification of A9 and A10 neurons”. This is not correct since some crucial papers identified A9 and A10 markers such GIRK2, SOX6, OTX2, Calbindin. Therefore these works should be cited (i.e. Panman et al., 2014; Di Salvio et al., 2010)
6) 6-OHDA is the correct abbreviation of 6-hydroxydopamine
7) VM organoids are cited but some important paper is missing (i.e. Qian et al., 2016, Cell)
8) Figure legends are very brief. Legends should clearly describe the content of the figures.
Author Response
The paper “Dopamine neuron diversity: recent advances and current challenges in human stem cell models and single cell sequencing” by Fiorenzano et al., shows an overview on the advances in the field of dopamine neuron modeling and genomics. The review is overall well organized and most of the appropriate works are covered.
We thank the Reviewer for their careful reading of our manuscript and constructive suggestions, which have helped us improve the quality of the paper. Please find below a point-by-point response to all comments.
Some improvements are still needed in order to be considered for publication as pointed out below:
1)The text needs some English language editing. Several sentences are too long and some spelling mistakes need to be fixed (i.e.” treat human brain disease(s)”)
The text has been edited for English language grammar, spelling, and fluency by a native English speaker.
2) Some references seem to be not consistent with the text where they are cited (i.e.Ref 27,28; 31-33)?
We have checked all the references and changed those that were inconsistent with the text.
3) The authors should cite some seminal work of Lorenz Studer group on dual/smad inhibition and ESC-derived dopamine neurons (i.e. Chambers et al., 2009; Kriks et al., 2011).
We have now added the missing references (page 4, line 142-143).
4) The authors should clarify that GSK3 inhibition is used in order to activate WNT pathway that contributes to dopamine neuron specification.
We agree with the Reviewer and have changed the sentence accordingly (page 4, line 140).
5) The authors state: “Although the absence of topological landmarks in in vitro prevents the precise identification of A9 and A10 neurons”. This is not correct since some crucial papers identified A9 and A10 markers such GIRK2, SOX6, OTX2, Calbindin. Therefore these works should be cited (i.e. Panman et al., 2014; Di Salvio et al., 2010).
We thank the Reviewer for raising this issue and for enabling us to rephrase and clarify this point. Although A9 and A10 neurons are formed in 2D culture, their architecture and spatial localization is lost. Such topological landmarks are normally used to distinguish A9 and A10 in endogenous human midbrain or in hPSC derived-grafts (on pag5 line 197).
6) 6-OHDA is the correct abbreviation of 6-hydroxydopamine
We have changed the abbreviation.
7) VM organoids are cited but some important paper is missing (i.e. Qian et al., 2016, Cell)
We have added the missing references on page 8 and line 338.
8) Figure legends are very brief. Legends should clearly describe the content of the figures.
We have now provided more extensive figure legends in order to better explain the illustrations.
Round 2
Reviewer 2 Report
The authors improved the manuscript in most of the raised points.
I have two still standing points:
1) Figure 3 still shows some spelling mistakes: "single cel; linage pathways"
2) I still don't see a logical text placement for the refs 34, 35, 36.
Author Response
We thank the Reviewer for their careful reading of our manuscript.
1) We have corrected spelling mistakes
2)We have changed the references to reflect the statements made in the text